# Parotid Space, a Different Space from Other Deep Neck Infection Spaces

**DOI:** 10.3390/microorganisms9112361

**Published:** 2021-11-15

**Authors:** Shih-Lung Chen, Chi-Kuang Young, Chun-Ta Liao, Tsung-You Tsai, Chung-Jan Kang, Shiang-Fu Huang

**Affiliations:** 1Department of Otorhinolaryngology & Head and Neck Surgery, Chang Gung Memorial Hospital, Linkou 333, Taiwan; rlong289@gmail.com (S.-L.C.); liaoct@cgmh.org.tw (C.-T.L.); mp0594@cgmh.org.tw (T.-Y.T.); handneck@gmail.com (C.-J.K.); 2School of Medicine, Chang Gung University, Taoyuan 333, Taiwan; rioriorioman@gmail.com; 3Department of Otorhinolaryngology, Chang Gung Memorial Hospital, Keelung 204, Taiwan; 4Department of Public Health, Chang Gung University, Taoyuan 333, Taiwan; 5Graduate Institute of Clinical Medical Sciences, Chang Gung University, Taoyuan 333, Taiwan; 6Department of Otorhinolaryngology & Head and Neck Surgery, Chang Gung Memorial Hospital at No. 5 Fu-Shin Street, Linkou 333, Taiwan

**Keywords:** deep neck infection, parotid space, *Klebsiella pneumoniae*, diabetes mellitus

## Abstract

Deep neck infections (DNIs) such as parotid abscesses are medical emergencies with a seemingly different etiology and treatment course from other DNIs. We sought to confirm this in the present retrospective population-based cohort study. Between August 2016 and January 2020, 412 patients with DNIs seen at a tertiary medical center were enrolled in this study. Infections of the parotid space were compared with those of other deep neck spaces, according to patient characteristics. All patients were divided into parotid space (PS; *n* = 91, 22.08%) and non-parotid space (NPS; *n* = 321, 77.92%) subgroups. We further divided the patients into single parotid space (PS-single; *n* = 50, 12.13%), single non-parotid space (NPS-single; *n* = 149, 36.16%), multiple parotid space (PS-multiple; *n* = 41, 9.95%), and multiple non-parotid space (NPS-multiple; *n* = 172, 41.76%) DNI subgroups. In the PS-single and PS-multiple subgroups, a longer duration of symptoms (*p* = 0.001), lower white blood cell count (*p* = 0.001), lower C-reactive protein level (*p* = 0.010), higher rate of ultrasonography-guided drainage (*p* < 0.001), and lower rates of surgical incision and drainage (*p* < 0.001) were observed compared with the NPS-single and NPS-multiple subgroups. The PS group had a higher positive *Klebsiella pneumoniae* culture rate (*p* < 0.001), and lower positive *Streptococcus constellatus* (*p* = 0.002), and *Streptococcus anginosus* (*p* = 0.025) culture rates than the NPS group. In a multivariate analysis, *K. pneumoniae* was independently associated with parotoid space involvement in comparisons of the PS and NPS groups, PS-single and NPS-single subgroups, and PS-multiple and NPS-multiple subgroups. The clinical presentation of a parotid space infection differs from that of other deep neck space infections.

## 1. Introduction

Deep neck infection (DNI) refers to infection of a space or fascial plane of the neck [1]. A DNI can be lethal and may also cause complications such as airway obstruction [2]. Treatment involves securing the airway, surgical- or ultrasonography (US)-guided drainage, adequate and prompt antimicrobial coverage, and appropriate management of complications [3]. Life-threatening complications of a DNI include carotid artery erosion, sepsis, disseminated intravascular coagulation, descending necrotizing mediastinitis, jugular vein thrombosis, necrotizing fasciitis, and airway obstruction [4]. DNIs continue to occur despite the widespread use of antibiotics [1].

The parotid space is relatively distant from other deep neck spaces. Most lymph nodes are located in the superficial parotid region or parotid tail [5,6]. The lymphatic drainage route in the parotid space is different from that in other spaces [7]. We hypothesized that microorganisms in the parotid space are different from those in other DNIs. Therefore, we performed this retrospective single-center study to compare infections between the parotid space and other deep neck spaces in terms of severity, clinical presentation, optimal treatment procedure, and infective pathogens.

## 2. Methods

By definition, DNIs occur in the fascial planes and spaces of the head and neck [8]. Diagnostic imaging modalities for DNIs include computed tomography (CT) with or without US. Patients meeting the clinical and radiological diagnostic criteria were enrolled in this study. In total, consecutive 595 patients were initially identified between August 2016 and January 2020. Patients with intraoral abscesses, peri-tonsillar abscesses, cervical necrotizing fasciitis, infections caused by penetrating or surgical neck trauma, severe cardiopulmonary disease, and/or an immunocompromised condition were excluded. Ultimately, 412 patients with a DNI who had been admitted to Chang Gung Memorial Hospital, Linkou, Taiwan, were enrolled. We retrospectively reviewed the medical records of the patients. Single- and multiple-space DNIs were distinguished based on CT scans. Incision and drainage were performed when the DNI compromised the airway, the abscess was large (≥2 cm), or if there was no improvement in the DNI after 48 h of intravenous empirical antibiotics.

The empiric antibiotics used in our hospital include ceftriaxone (1 g, q12h) and metronidazole (500 mg, q8h). The antibiotic regime may be changed according to cultured pathogen and antibiotic susceptibility test results. If no causative microorganisms are identified, patients receive intravenous antibiotic treatment for 7–10 days. Then, we switched from an intravenous to an oral form of amoxicillin trihydrate + clavulanate potassium, clindamycin, or cephalosporin to complete the treatment course (i.e., intravenous antibiotics for 7 days and oral antibiotics for another 7 days).

### 2.1. Statistical Analysis

Patient demographics such as sex and age (range: 2–98 years), and medical data such as duration of symptoms (range: 1–30 days), length of hospital stay (range: 2–52 days), white blood cell (WBC) count (range: 1600–42,700 uL), C-reactive protein (CRP) level (range: 4.3–487 mg/L), blood glucose level (range: 61–564 mg/dL), diabetes mellitus (DM) status, the performance of incision or drainage surgery, US-guided drainage status, the performance of tracheostomy, and cultured pathogens were retrospectively retrieved from the medical records of our hospital. For continuous variables, the data are presented as the mean ± standard deviation; categorical variables are presented as numbers (%). The data were analyzed using MedCalc software (ver. 18.6; MedCalc, Ostend, Belgium). A preliminary Kolmogorov–Smirnov test revealed that the data were not normally distributed. Therefore, we used Fisher’s exact test, the Mann–Whitney U test, and the Kruskal–Wallis test to analyze the data. Univariate and multivariate logistic regressions were performed using forward stepwise selection, and all variables included in the univariate analysis were entered into the final multivariate model. A *p*-value < 0.05 was considered statistically significant.

### 2.2. Ethics Statement

This study was approved by the Institutional Review Board (IRB) of the Chang Gung Medical Foundation (IRB no. 202000619B0). The data were collected retrospectively and anonymized before analysis. The IRB waived the requirement for informed consent.

## 3. Results

We included 278 males (67.47%) and 134 females (32.53%), with a mean age of 51.14 ± 18.84 years (Table 1). The duration of symptoms was 5.01 ± 4.51 days, and the mean hospital stay was 9.48 ± 8.01 days. The mean WBC count was 15,059.94 ± 5916.45/µL, while the mean CRP level was 137.95 ± 107.97 mg/L, and the mean blood glucose level was 142.47 ± 72.10 mg/dL. A total of 156 patients had DM (37.86%). In addition, 188 patients underwent incisional and drainage surgery (45.63%), and 72 underwent US-guided drainage surgery (17.47%).

All patients were divided into parotid space (PS; *n* = 91, 22.08%) and non-parotid space (NPS; *n* = 321, 77.92%) DNI groups. The patients were further divided into the following four subgroups: infection of a single parotid space (PS-single; *n* = 50, 12.13%), infection of multiple spaces, including a parotid space (PS-multiple; *n* = 41, 9.95%), infection of a single non-parotid space (NPS-single; *n* = 149, 36.16%), and infection of multiple non-parotid spaces (NPS-multiple; *n* = 172, 41.76%). A total of 44 patients (10.67%) underwent tracheostomy. The frequency rates for pathogens affecting more than 2% of all patients are listed in Table 1. The pathogens in 94 patients (22.81%) were polymicrobial.

Table 2 shows the clinical data for the PS and NPS groups. The former group had a longer duration of symptoms (*p* = 0.001), lower WBC count (*p* = 0.001), lower CRP level (*p* = 0.010), higher rate of US-guided drainage (*p* < 0.001), lower rates of surgical incision and drainage (*p* < 0.001), higher *Klebsiella pneumoniae* culture rate (*p* < 0.001), and lower *Streptococcus constellatus* (*p* = 0.002) and *Streptococcus anginosus* (*p* = 0.025) culture rates. The hospital stay, glucose level, DM status, and tracheostomy rate did not differ between the PS and NPS groups. In our multivariate analysis, the duration of symptoms, surgical incision and drainage, US-guided drainage, and *K. pneumoniae* were independently associated with parotoid space involvement.

Table 3 shows the data for the PS-single and NPS-single subgroups. The duration of symptoms was 7.00 ± 6.18 and 4.90 ± 4.29 days in the PS-single and NPS-single subgroups, respectively (*p* = 0.031); the respective WBC counts were 11,630.00 ± 3867.19/µL and 13,932.20 ± 4990.01/µL (*p* = 0.001), and the respective CRP levels were 60.56 ± 68.83 and 94.94 ± 84.54 mg/L (*p* < 0.001). Regarding treatments, 6.00% of the PS-single patients underwent surgical incision and drainage, compared with 44.29% of the NPS-single patients (*p* < 0.001). Meanwhile, 44.00% of the PS-single patients underwent US-guided drainage, compared with 14.09% of the NPS-single patients (*p* < 0.001). Furthermore, *K. pneumoniae* infection was more common in the PS-single (46.00%) than in the NPS-single (2.68%) group (*p* < 0.001), while *S. constellatus* infection was less common in the PS-single (0.00%) than in the NPS-single (10.73%) group (*p* = 0.013). In our multivariate analysis, surgical incision and drainage, US-guided drainage, and *K. pneumoniae* were independently associated with parotoid space involvement.

Table 4 shows the data for the PS-multiple and NPS-multiple subgroups. The duration of symptoms was 6.65 ± 6.80 and 4.11 ± 2.88 days in the PS-multiple and NPS-multiple subgroups, respectively (*p* = 0.016). Surgical incision and drainage were less common in the PS-multiple group (39.02%) than in the NPS-multiple (59.88%) group (*p* = 0.022), while PS-multiple patients (24.39%) were more likely to undergo US-guided drainage than NPS-multiple patients (11.05%) (*p* = 0.039). The proportion of *K. pneumoniae* infections was significantly higher in the PS-multiple group (48.78%) than in the NPS-multiple group (5.23%) (*p* < 0.001). In our multivariate analysis, the time durations of symptoms, US-guided drainage, and *K. pneumoniae* were independently associated with parotoid space involvement.

Figure 1 shows the laboratory data for the four subgroups. Figure 1A shows the mean WBC counts of the PS-single, NPS-single, PS-multiple, and NPS-multiple subgroups, which were 11,695.91 ± 3878.78/µL, 13,932.21 ± 4990.01/µL, 15,748.78 ± 6800.92/µL, and 16,869.77 ± 6286.58/µL, respectively (*p* < 0.001). Notably, the WBC count of the PS-single subgroup differed significantly from that of the other three subgroups; the WBC count also differed significantly between the NPS-single and NPS-multiple subgroups (*p* < 0.05; light blue line).

Figure 1B shows the mean CRP levels of the PS-single, NPS-single, PS-multiple, and NPS-multiple subgroups, which were 60.88 ± 69.50, 94.56 ± 84.71, 192.59 ± 119.58, and 184.51 ± 104.79 mg/L, respectively (*p* < 0.001). The mean CRP levels of the PS-single and PS-multiple subgroups differed significantly, as did those of the NPS-single and NPS-multiple subgroups.

## 4. Discussion

We found significant differences between DNIs of the parotid space and other deep neck spaces. Table 1 shows that 67.47% and 32.53%; of all DNIs affected males and females, respectively, similar to a previous report showing male predominance [9,10]. Most of our patients were middle aged (mean age, 51.14 ± 18.84 years), as in previous DNI studies [10,11].

Table 1 shows that the top two pathogens were *K. pneumoniae* (13.59%) and *S. constellatus* (11.89%). The culture rates of *K. pneumoniae* (an anaerobic Gram-negative bacterium) were high in patients with parotid space involvement, at 46% and 48.78% in the PS-single and PS-multiple subgroups, respectively. This result is in line with Chi et al. and Huang et al. [1,12]. The latter study reported a parotid space culture rate of up to 58.3%. In contrast, *S. constellatus*, an aerobic Gram-positive bacterium, was more common in our NPS patients, with rates of 13.44% and 17.44% in the NPS-single and NPS-multiple subgroups, respectively; the respective *K. pneumoniae* culture rates were only 2.68% and 5.23%. As parotid space infections are usually caused by *K. pneumoniae*, Huang et al. suggested that the route of parotid entry may differ from that of other spaces. Most DNIs are of dental and upper airway origin [10,13]. However, parotid space infections are often caused by sialadenitis, sialoliths, parotid duct obstruction, lymphadenitis, and skin infections [2,9,14].

In this study, *K. pneumoniae* was independently associated with parotid space involvement. The human virulence factors for *K. pneumoniae* include a biofilm, a capsule, lipopolysaccharide, and siderophores that overcome mechanical and chemical barriers, as well as humoral and cellular innate defenses, to establish infection [15,16]. *Klebsiella pneumoniae* causes serious infections in immunocompromised individuals, as well as in those who are healthy and immunocompetent.

Ganesh et al. and Lee et al. reported that *Staphylococcus aureus* was the most common parotid space pathogen in Singapore [17,18]. Therefore, pathogens may vary geographically. *K. pneumoniae* was still the second most common pathogen in the study of Lee et al. Wang et al. proposed that the various antibiotics used before admission might affect the flora, as reflected in varying culture results [13]. Additionally, the microbiology of DNIs changes over time [1]. The pathogens cultured from the parotid space varied among studies, and we also consider a DNI of the parotid space to differ from those of other spaces.

Table 2 shows that PS patients generally had a longer duration of symptoms than NPS patients (*p* = 0.001). Similar findings are shown in Table 3 (PS-single vs. NPS-single, *p* = 0.031; PS-multiple vs. NPS-multiple, *p* = 0.016). Patients with parotid space involvement had less severe symptoms than those with non-parotid space involvement. Table 2 also shows that the treatment modalities differed between the PS and NPS groups (*p* < 0.001). US-guided aspiration was used most frequently in PS patients, while surgical incision and drainage were more common in NPS patients. As the parotid space is relatively superficial, US-guided drainage, which is efficient and less invasive than other modalities, can be used; this also preserves the facial nerve.

Figure 1A,B show the mean WBC counts and CPR levels, respectively; both parameters had higher values in multiple-space infections, indicating that these were more severe than single-space infections.

DM was the most common systemic disease among those DNI patients who had a severe clinical course and poor outcome [1,11,12,19,20]. DM reduces immunity, impairs neutrophil function and complement activation, and increases susceptibility to infection. Chi et al. reported that DM predisposed patients to parotid space abscesses [21]. As shown in Table 2, the glucose levels and DM incidence did not differ significantly between the PS and NPS groups. Similarly, as shown in Table 3, the blood glucose levels of the PS-single (149.88 mg/dL) and PS-multiple (161.53 mg/dL) subgroups were higher than those of the NPS-single (124.30 mg/dL) and NPS-multiple (151.52 mg/dL) subgroups, although the differences were not significant. DM is associated with xerostomia, hyposalivation, and salivary gland diseases [22,23]. Blood glucose control is usually considered important for infection control in DNIs patients, especially in cases of PS infection (because fluctuations in the blood glucose level can exacerbate a parotid infection due to salivary flow stasis or xerostomia).

Previous studies demonstrated that, among DM patients, the proportion of *K. pneumoniae* DNIs is high [1,11]. In this study, the incidence rates of *K. pneumoniae* DNIs in the PS-single and NPS-single subgroups were 46% and 2.68%, respectively (*p* < 0.001), while those in the PS-multiple and NPS-multiple subgroups were 48.78% and 5.22%, respectively (*p* < 0.001). However, the incidence of DM did not differ between the PS-single and NPS-single (*p* = 0.910) subgroups, or between the PS-multiple and NPS-multiple (*p* = 0.415) subgroups (Table 3). A correlation existed between parotid space abscesses and *K. pneumoniae* culture rates but not between parotid space abscesses and DM status.

### Limitations

To the best of our knowledge, this is the first retrospective review of a large cohort to discuss differences between infections of the parotid space and other deep neck spaces. However, the retrospective design gave rise to a high attrition rate. Moreover, the sizes of the four subgroups differed. The NPS-single and NPS-multiple subgroups included the majority of the study population, potentially giving rise to selection bias.

## 5. Conclusions

Parotid space abscess was associated with a longer duration of symptoms, lower WBC count and CRP level, higher rate of US-guided drainage, lower rate of surgical incision and drainage, higher prevalence of *K. pneumoniae*, and lower prevalence of *S. constellatus* and *S. anginosus* compared with DNIs in other spaces. These results can be explained by the differences in location and route of bacterial infection between the parotoid space and other deep neck spaces. The parotid space is a unique deep neck space; infections therein should thus be treated differently from those in other spaces in clinical practice.

## Figures and Tables

**Figure 1 microorganisms-09-02361-f001:**
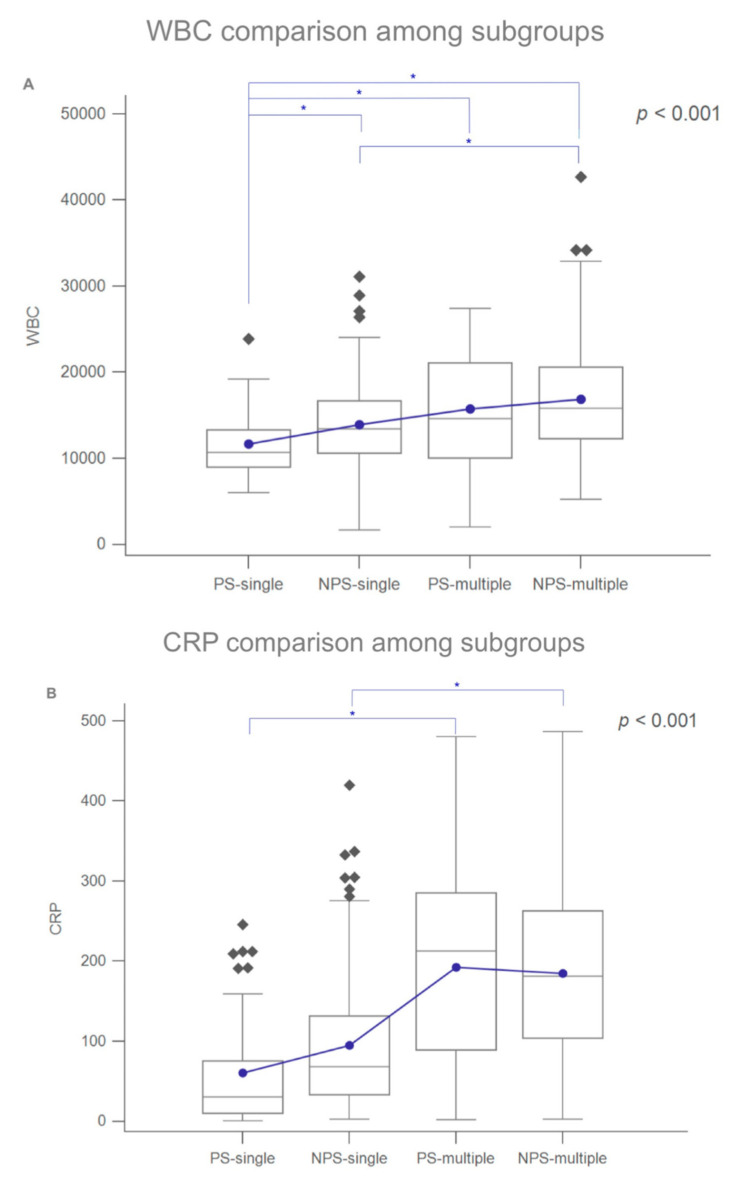
(**A**) Box-and-whisker plots of the laboratory data for the four subgroups. WBC counts of the subgroups. (**B**) CRP levels of the subgroups. Blue dots: means; diamonds: mild outliers; squares: extreme outliers. Blue lines/asterisks indicate statistical significance (*p* < 0.05).

**Table 1 microorganisms-09-02361-t001:** Clinicopathological characteristics of the 412 patients.

Characteristics	Values
Males, N (%)	278 (67.47)
Females, N (%)	134 (32.53)
Age, years, mean ± SD	51.14 ± 18.84
Duration of symptoms, days, mean ± SD	5.01 ± 4.51
Hospital stay, days, mean ± SD	9.48 ± 8.01
WBC count, μL, mean ± SD	15,059.94 ± 5916.45
CRP, mg/L, mean ± SD	137.95 ± 107.97
Glucose, mg/dL, mean ± SD	142.47 ± 72.10
Diabetes mellitus, N (%)	156 (37.86)
Surgical incision and drainage, N (%)	188 (45.63)
Ultrasonography-guided drainage, N (%)	72 (17.47)
Deep neck infection involving PS, N (%)	91 (22.08)
PS-single, N (%)	50 (12.13)
PS-multiple, N (%)	41 (9.95)
Deep neck infection involving NPS, N (%)	321 (77.92)
NPS-single, N (%)	149 (36.16)
NPS-multiple, N (%)	172 (41.76)
Tracheostomy, N (%)	44 (10.67)
Culture pathogens	
*Klebsiella pneumoniae*, N (%)	56 (13.59)
*Streptococcus constellatus*, N (%)	49 (11.89)
*Streptococcus anginosus*, N (%)	31 (7.52)
*Parvimonas micra*, N (%)	29 (7.03)
*Prevotella buccae*, N (%)	27 (6.55)
*Prevotella intermedia*, N (%)	18 (4.36)
*Staphylococcus aureus*, N (%)	12 (2.91)
Other species, N (%)	117 (28.39)
No growth, N (%)	167 (40.57)

N, number; SD, standard deviation; WBC, white blood cell (normal range: 3500–11,000/μL); CRP, C-reactive protein (normal range: <5 mg/L); PS, parotid space; NPS, non-parotid space. Glucose normal range: 70–100 mg/dL; other species = pathogens seen in <2% of the patients.

**Table 2 microorganisms-09-02361-t002:** Clinical data from the PS and NPS deep neck infection subgroups.

Characteristics		Univariate Analysis			Multivariate Analysis	
	PS	NPS	*p*-Value	OR	CI	*p*-Value
Total, N (%)	91 (100.0)	321 (100.0)				
Males, N (%)	67 (73.62)	211 (65.73)	0.165			
Females, N (%)	24 (26.38)	110 (34.27)				
Age, years, mean ± SD	52.09 ± 19.56	50.87 ± 18.65	0.544			
Duration of symptoms, days, mean ± SD	6.84 ± 6.43	4.47 ± 3.62	**0.001 ***	1.072	1.013–1.134	**0.014 ***
Hospital stay, days, mean ± SD	11.17 ± 9.70	8.97 ± 7.39	0.082			
WBC count, μL, mean ± SD	13,485.71 ± 5739.79	15,506.23 ± 5898.16	**0.001 ***	-	-	-
CRP, mg/L, mean ± SD	120.95 ± 115.31	143.03 ± 105.43	**0.010 ***	-	-	-
Glucose, mg/dL, mean ± SD	155.13 ± 97.25	138.88 ± 62.92	0.423			
Diabetes mellitus			0.393			
Yes, N (%)	38 (41.76)	118 (36.76)				
No, N (%)	53 (58.24)	203 (63.24)				
Surgical incision and drainage			<**0.001 ***			**0.005 ***
Yes, N (%)	19 (20.88)	169 (52.64)		0.384	0.195–0.753	
No, N (%)	72 (79.12)	152 (47.36)		1.000		
Ultrasonography-guided drainage			<**0.001 ***			**0.001 ***
Yes, N (%)	32 (35.17)	40 (12.47)		3.267	1.621–6.583	
No, N (%)	59 (64.83)	281 (87.53)		1.000		
Tracheostomy			0.083			
Yes, N (%)	5 (5.49)	39 (12.14)				
No, N (%)	86 (94.51)	282 (87.86)				
Culture pathogens						
*Klebsiella pneumoniae*, N (%)	43 (47.25)	13 (4.04)	<**0.001 ***	21.885	10.428–45.932	<**0.001 ***
*Streptococcus constellatus*, N (%)	3 (3.29)	46 (14.33)	**0.002 ***	-	-	-
*Streptococcus anginosus*, N (%)	2 (2.19)	29 (9.03)	**0.025 ***	-	-	-
*Parvimonas micra*, N (%)	5 (5.49)	24 (7.47)	0.645			
*Prevotella buccae*, N (%)	8 (8.79)	19 (5.91)	0.339			
*Prevotella intermedia*, N (%)	3 (3.29)	15 (4.67)	0.773			
*Staphylococcus aureus*, N (%)	5 (5.49)	7 (2.18)	0.141			

PS, deep neck infection involving the parotid space; NPS, deep neck infection not involving the parotid space; N, number; SD, standard deviation; WBC, white blood cell (normal range: 3500–11,000/μL); CRP, C-reactive protein (normal range: <5 mg/L); OR, odds ratio; CI, confidence interval. Glucose normal range: 70–100 mg/dL); * *p* < 0.05 (significant differences are shown in bold).

**Table 3 microorganisms-09-02361-t003:** Clinical data from the PS-single and NPS-single deep neck infection subgroups.

Characteristics		Univariate Analysis			Multivariate Analysis	
	PS-Single	NPS-Single	*p*-Value	OR	CI	*p*-Value
Total, N (%)	50 (100.0)	149 (100.0)				
Males, N (%)	37 (74.00)	106 (71.14)	0.855			
Females, N (%)	13 (26.00)	43 (28.86)				
Age, years, mean ± SD	49.72 ± 19.31	48.30 ± 17.54	0.665			
Duration of symptoms, days, mean ± SD	7.00 ± 6.18	4.90 ± 4.29	**0.031 ***	-	-	-
Hospital stay, days, mean ± SD	9.00 ± 9.05	6.69 ± 5.78	0.265			
WBC count, μL, mean ± SD	11,630.00 ± 3867.19	13,932.20 ± 4990.01	**0.001 ***	-	-	-
CRP, mg/L, mean ± SD	60.56 ± 68.83	94.94 ± 84.54	<**0.001 ***	-	-	-
Glucose, mg/dL, mean ± SD	149.88 ± 92.83	124.30 ± 42.04	0.910			
Diabetes mellitus			0.596			
Yes, N (%)	17 (34.00)	44 (29.53)				
No, N (%)	33 (66.00)	105 (70.47)				
Surgical incision and drainage			<**0.001 ***			**0.058 ***
Yes, N (%)	3 (6.00)	66 (44.29)		0.073	0.011–0.468	
No, N (%)	47 (94.00)	83 (55.71)		1.000		
Ultrasonography-guided drainage			<**0.001 ***			**0.024 ***
Yes, N (%)	22 (44.00)	21 (14.09)		3.096	1.155–8.296	
No, N (%)	28 (56.00)	128 (85.91)		1.000		
Tracheostomy			0.068			
Yes, N (%)	0 (0.00)	10 (16.71)				
No, N (%)	50 (100.0)	139 (93.29)				
Culture pathogens						
*Klebsiella pneumoniae*, N (%)	23 (46.00)	4 (2.68)	<**0.001 ***	62.796	12.687–310.808	<**0.001 ***
*Streptococcus constellatus*, N (%)	0 (0.00)	16 (10.73)	**0.013 ***	-	-	-
*Streptococcus anginosus*, N (%)	1 (2.00)	11 (7.38)	0.301			
*Parvimonas micra*, N (%)	2 (4.00)	13 (8.72)	0.364			
*Prevotella buccae*, N (%)	0 (0.00)	5 (3.35)	0.333			
*Prevotella intermedia*, N (%)	0 (0.00)	7 (4.69)	0.195			
*Staphylococcus aureus*, N (%)	4 (8.00)	3 (2.01)	0.068			

PS-single, infection of a single parotid space; NPS-single, infection of a single space other than the parotid space; N, number; SD, standard deviation; WBC, white blood cell count (normal range: 3500–11,000/μL); CRP, C-reactive protein (normal range: <5 mg/L); OR, odds ratio; CI, confidence interval. Glucose normal range: 70–100 mg/dL; * *p* < 0.05 (significant differences are shown in bold).

**Table 4 microorganisms-09-02361-t004:** Clinical data from the PS-multiple and NPS-multiple deep neck infection subgroups.

Characteristics		Univariate Analysis			Multivariate Analysis	
	PS-Multiple	NPS-Multiple	*p*-Value	OR	CI	*p*-Value
Total, N (%)	41 (100.0)	172 (100.0)				
Males, N (%)	30 (73.17)	105 (61.04)	0.206			
Females, N (%)	11 (26.83)	67 (38.96)				
Age, years, mean ± SD	55.00 ± 19.72	53.11 ± 19.34	0.498			
Duration of symptoms, days, mean ± SD	6.65 ± 6.80	4.11 ± 2.88	**0.016 ***	1.102	1.005–1.209	**0.037 ***
Hospital stay, days, mean ± SD	13.82 ± 9.91	10.95 ± 8.06	0.083			
WBC, μL, mean ± SD	15,748.78 ± 6800.92	16,869.77 ± 6286.58	0.402			
CRP, mg/L, mean ± SD	192.59 ± 119.58	184.69 ± 104.19	0.710			
Glucose, mg/dL, mean ± SD	161.53 ± 103.18	151.52 ± 74.32	0.415			
Diabetes mellitus			0.384			
Yes, N (%)	21 (51.21)	74 (43.02)				
No, N (%)	20 (48.79)	98 (56.98)				
Surgical incision and drainage			**0.022 ***	-	-	-
Yes, N (%)	16 (39.02)	103 (59.88)				
No, N (%)	25 (60.98)	69 (40.12)				
Ultrasonography-guided drainage			**0.039 ***			**0.033 ***
Yes, N (%)	10 (24.39)	19 (11.05)		3.169	1.093–9.188	
No, N (%)	31 (75.61)	153 (88.95)		1.000		
Tracheostomy			0.635			
Yes, N (%)	5 (12.19)	29 (16.86)				
No, N (%)	36 (87.81)	143 (83.14)				
Culture pathogens						
*Klebsiella pneumoniae*, N (%)	20 (48.78)	9 (5.23)	<**0.001 ***	16.677	6.462–43.034	<**0.001 ***
*Streptococcus constellatus*, N (%)	3 (7.31)	30 (17.44)	0.148			
*Streptococcus anginosus*, N (%)	1 (2.43)	18 (10.46)	0.133			
*Parvimonas micra*, N (%)	3 (7.31)	11 (6.39)	0.735			
*Prevotella buccae*, N (%)	8 (19.51)	14 (8.13)	0.139			
*Prevotella intermedia*, N (%)	3 (7.31)	8 (4.65)	0.446			
*Staphylococcus aureus*, N (%)	1 (2.43)	4 (2.32)	1.000			

PS-multiple, infection of multiple spaces including the parotid space; NPS-multiple, infection of multiple spaces not including the parotid space; N, numbers; SD, standard deviation; WBC, white blood cell count (normal range: 3500–11,000/μL); CRP, C-reactive protein (normal range: <5 mg/L); OR, odds ratio; CI, confidence intervals. Glucose normal range: 70–100 mg/dL; * *p* < 0.05 (significant differences are shown in bold).

## Data Availability

All data generated or analyzed during this study are included in this published article. The data are available on request.

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
