# Peer review of "Parotid Space, a Different Space from Other Deep Neck Infection Spaces"

_microorganisms, 2021, doi:10.3390/microorganisms9112361_

Round 1
Reviewer 1 Report
Thank you for giving me the opportunity to review this work.
The authors included 412 patients which is quite a big cohort on DNI with 91 patients with parotid space infections.
Unfortunately, the paper lacks accuracy and could be significantly improved in term of writing, English language and statistical analysis. It is hard to consider an original work without a multivariate analysis.
Hoping my comment would help you improve your work.
INTRODUCTION
Major comment :
Please provide further references
Minor comments :
1) A DNI can be lethal, and may also cause complications or airway obstruction : please replace by "A DNI can be lethal, and may also cause complications AS airway obstruction".
2) Treatment involves securing the airway, providing adequate antimicrobial coverage, sur- gical drainage, ultrasonography(US)-guided drainage, and appropriate management of complications : I would modify as follows "Treatment involves securing the airway, providing surgical drainage OR ultrasonography(US)-guided drainage, adequate AND PROMPT antimicrobial coverage, and appropriate management of complications.
3) In the same sentence "...and appropriate management of complications" : which complications ?
4) "Despite the widespread use of antibiotics, a DNI remains one of the most difficult emergencies encountered in daily clinical practice" : What is the link between the wide use of antibiotic and the diagnosis of DNI ?
DNI appears despite the wide use of antibiotics makes more sense to me!
5) "Lymphatic drainage of the parotid space differs from that of the other spaces. The lymphatic drainage route in the parotid space is different from other spaces." : Please delete one sentence.
6) We speculated that the behaviors and pathogens in parotid space could be different from other deep neck infection spaces ==> We speculated that the behaviors and microorganisms found in the parotid space are different from those of other deep neck infection spaces.
7) Thus, we hypothesized that the parotid space infection is different from other DNIs : this sentence is repetitive compared to the previous one. Please delete this sentence or modify.
8) Line 46 : I would suggest to modify as follows : “Therefore, we performed a retrospective single center study to compare infections from the parotid space to other deep neck spaces …”.
Methods
Major comment :
- Please provide more details about patient’s inclusion criteria.
- Please provide a clear definition of DNIs
- Were all patients consecutively admitted to the hospital included ?
- Please reorganize the section with a dedicate paragraph for definition, and one concerning the management of DNIs in your institution.
- The statistical analysis is poor. It could be improved.
Minor comments
- Please provide details when you use an abbreviation : US line 53
- Line 62 : the type of treatment should be included in the results parts or in the table with numbers and not in the method section :” Treatment included antibiotics, US-guided needle drainage, and surgical incision and drainage».
- The number of patients included in each group should also be included in the results and not in the method section.
- Please rephrase the sentence line 64. For instance : “Incision and drainage were performed in case of : ….”
- Line 69 : you provide the details on the empiric antibiotic therapy, consequently no need to inform the reader that these antibiotic were used before the cultures were available as it is the definition of empiric antibiotic regimen.
- Please use appropriate words concerning the reassessment of antibiotic therapy which was tailored to culture and susceptibility data.
- Please provide the duration of the complete treatment course (line 77).
- Exclusion criteria should be included in the first paragraph of the method section.
- Please reorganize the statistical analysis section with
Patients’ demographic (gender…) and medical data (…) were retrospectively retrieved from ….
- The mean ± standard deviation was used for continuous variables. What about categorical variables ?
Results :
Major comment
The result section lacks a multivariate analysis.
Minor comment :
- Please start the result section with :
- We included … patients … with …
Here you should add the data previously included in the method section.
- Please verify this expression : The chief compliant period ? Do you meet the duration of symptoms?
- Please detail the abbreviation DM ?
Discussion:
I would suggest to further discuss your results and then add more references.
Reviewer 2 Report
It is an interesting study on a large cohort of patients. Ethical considerations and ethical approval number should be mentioned.
The introductory part is pretty short. I think it could be developed and some more references should be added.
In the discussion part, I would recommend more comments on the therapeutic options with appropiate citations.
